# VEGFA Status as a Predictive Marker of Therapy Outcome in Metastatic Gastric Cancer Patients Following Ramucirumab-Based Treatment

**DOI:** 10.3390/biomedicines11102721

**Published:** 2023-10-07

**Authors:** Annalisa Schirizzi, Aram Arshadi, Doron Tolomeo, Laura Schirosi, Anna Maria Valentini, Giampiero De Leonardis, Maria Grazia Refolo, Rossella Donghia, Clelia Tiziana Storlazzi, Alfredo Zito, Angela Dalia Ricci, Simona Vallarelli, Carmela Ostuni, Maria Bencivenga, Giovanni De Manzoni, Caterina Messa, Raffaele Armentano, Gianluigi Giannelli, Claudio Lotesoriere, Rosalba D’Alessandro

**Affiliations:** 1Laboratory of Experimental Oncology, National Institute of Gastroenterology, IRCCS “S. de Bellis” Research Hospital, 70013 Castellana Grotte, Italy; annalisa.schirizzi@irccsdebellis.it (A.S.); giampiero.deleonardis@irccsdebellis.it (G.D.L.); caterina.messa@irccsdebellis.it (C.M.); 2Department of Biosciences, Biotechnology and Environment, University of Bari Aldo Moro, 70125 Bari, Italy; arshadia89@gmail.com (A.A.); doron.tolomeo@uniba.it (D.T.); cleliatiziana.storlazzi@uniba.it (C.T.S.); 3Pathology Department, IRCCS Istituto Tumori “Giovanni Paolo II”, 70124 Bari, Italy; l.schirosi@oncologico.bari.it (L.S.); a.zito@oncologico.bari.it (A.Z.); 4Histopathology Unit, National Institute of Gastroenterology, IRCCS “S. de Bellis” Research Hospital, 70013 Castellana Grotte, Italy; am.valentini@irccsdebellis.it (A.M.V.); raffaele.armentano@irccsdebellis.it (R.A.); 5Clinical Pathology Unit, National Institute of Gastroenterology, IRCCS “S. de Bellis” Research Hospital, 70013 Castellana Grotte, Italy; maria.refolo@irccsdebellis.it; 6Data Science Unit, National Institute of Gastroenterology, IRCCS “S. de Bellis” Research Hospital, 70013 Castellana Grotte, Italy; rossella.donghia@irccsdebellis.it; 7Medical Oncology Unit, National Institute of Gastroenterology, IRCCS “S. de Bellis” Research Hospital, 70013 Castellana Grotte, Italy; angela.ricci@irccsdebellis.it (A.D.R.); simona.vallarelli@irccsdebellis.it (S.V.); carmela.ostuni@irccsdebellis.it (C.O.); 8General and Upper GI Surgery Division, University of Verona, 37126 Verona, Italy; maria.bencivenga@univr.it (M.B.); giovanni.demanzoni@univr.it (G.D.M.); 9Scientific Direction, National Institute of Gastroenterology, IRCCS “S. de Bellis” Research Hospital, 70013 Castellana Grotte, Italy; gianluigi.giannelli@irccsdebellis.it

**Keywords:** metastatic Gastric Cancer, VEGFA, anti-angiogenic therapy, predictive marker, micro-vessels

## Abstract

Metastatic gastric cancer (mGC) often has a poor prognosis and may benefit from a few targeted therapies. Ramucirumab-based anti-angiogenic therapy targeting the VEGFR2 represents a milestone in the second-line treatment of mGC. Several studies on different cancers are focusing on the major VEGFR2 ligand status, meaning VEGFA gene copy number and protein overexpression, as a prognostic marker and predictor of response to anti-angiogenic therapy. Following this insight, our study aims to examine the role of VEGFA status as a predictive biomarker for the outcome of second-line therapy with Ramucirumab and paclitaxel in mGC patients. To this purpose, the copy number of the *VEGFA* gene, by fluorescence in situ hybridization experiments, and its expression in tumor tissue as well as the density of micro-vessels, by immunohistochemistry experiments, were assessed in samples derived from mGC patients. This analysis found that amplification of *VEGFA* concomitantly with VEGFA overexpression and overexpression of VEGFA with micro-vessels density are more represented in patients showing disease control during treatment with Ramucirumab. In addition, in the analyzed series, it was found that amplification was not always associated with overexpression of VEGFA, but overexpression of VEGFA correlates with high micro-vessel density. In conclusion, overexpression of VEGFA could emerge as a potential biomarker to predict the response to anti-angiogenic therapy.

## 1. Introduction

Gastric cancer (GC) is the fifth most frequent malignancy in people over 70 years of age and the fifth leading cause of cancer death in the general population. In Western countries, 80% of patients are diagnosed with advanced unresectable disease or relapse within 5 years after curative surgery. Therefore, the prognosis of advanced GC remains poor, with a 5-year survival rate of <30% for all stages and <4% for metastatic disease [1,2]. The search for predictive/prognostic markers assumes a crucial role in the treatment of locally advanced and metastatic GC (mGC). Despite the advent of immunotherapy and new molecularly targeted therapies, the therapeutic options available for these patients are limited to date [3].

Similar to other neoplastic diseases, the treatment of GC may benefit from anti-angiogenic drugs including Ramucirumab, a monoclonal antibody antagonist of vascular endothelial growth factor receptor-2 (VEGFR-2), currently used in the second-line therapy of mGC and gastro-esophageal junction carcinomas. Based on the results of two different randomized phase III trials, Ramucirumab could be used as monotherapy or in combination with paclitaxel (PTX) in pre-treated patients after platinum- and fluoropyrimidine-based therapy [4,5].

Great interest is currently focused on the study of growth factors belonging to the VEGF family and their receptor tyrosine kinases that mediate pro-angiogenetic effects with a key role in the pathogenesis of gastric carcinoma. These are structurally related biological molecules that include: VEGFA, VEGFB, VEGFC, VEGFD, and placental growth factor (PlGF) [6,7]. VEGFA is the major member of this family and performs its function mainly through binding to VEGFR2 via auto-phosphorylation mechanisms that activate downstream signal pathways involved in endothelial cell (ECs) proliferation, survival and motility playing a key role in physiological and pathological angiogenesis [8]. VEGFA overexpression in the tumor represents mainly the response to increased tumor hypoxia via the hypoxia-inducible factor-1α (HIF-1α) pathway [9]. Moreover, VEGF plays a crucial role in tumor growth, even in an angiogenesis-independent way, by interacting with receptors expressed on tumor cells through autocrine and/or paracrine mechanisms [10].

Furthermore, members of the VEGF family are involved at multiple levels in the fine-tuning of the cancer immune cycle, resulting in substantial changes that contribute to a microenvironment enabling the tumor to evade immune surveillance. The angiogenic and immune profiles correlate with the molecular classes of GC. The microsatellite instability (MSI) and Epstein–Barr virus (EBV) positive subtypes are characterized by high programmed cell death ligand 1 (PD-L1) expression, a cytotoxic lymphocyte infiltrate rich in CD8+ TILs, and a high mutational rate (tumor mutational burden ≥10); these molecular classes of GC have been referred to as ‘hot tumors’ that are susceptible to immunotherapeutic drugs. In contrast, molecular subgroups of genetically stable (GS) tumors or with chromosomal instability (CIN) are referred to as ‘cold tumors’ and are often characterized by an immunosuppressive microenvironment containing factors that promote angiogenesis [11]. The CIN group includes tumors with *VEGFA* amplification, an alteration found in GC with an incidence of 7% [12,13].

In the human genome, the *VEGFA* gene is located on the short arm of chromosome 6 (chr6), at position p12-21 [12]. Mutations involving copy number alterations of the *VEGFA* gene could account for its overexpression in tumor tissue.

To select patients eligible for targeted therapies, copy number (polysomy or amplification) assessment of the target gene, measured by fluorescent in situ hybridization (FISH), and/or overexpression, measured at the protein level by immunohistochemistry (IHC), is often required. The concordance between gene locus status and overexpression of the corresponding protein is still widely debated [14].

Several studies on *HER2* have led to the definition of gene amplification as a marker of positive response to trastuzumab, a monoclonal antibody that interferes with the HER2 receptor. In breast cancer and GC, the concordance between amplification and overexpression is very high when the protein overexpression score is high, while a discrepancy can be detected in moderate overexpression cases. The guidelines recommend performing in situ hybridization (ISH) in the latter group of patients before addressing trastuzumab therapy [15,16,17].

Recently, the impact of VEGFA status (gene copy number and protein overexpression) has been under investigation for some cancer types, such as osteosarcoma, colorectal, breast, and liver cancer [18,19,20,21]. Studies conducted in hepatocellular carcinoma (HCC) reported that patients with *VEGFA* amplification were particularly responsive to VEGFA-targeted therapy, and sorafenib specifically, suggesting that it can be used as a predictive biomarker for personalized treatment of HCC with sorafenib [19,21,22,23]. Studies performed on the detection of VEGFA status in osteosarcoma revealed that both FISH and IHC provide additional benefits in the eventual selection of patients candidates for anti-angiogenic treatment, whereas evaluation of only one of the two parameters proved to be insufficient or misleading [18]. The VERA trial, through validation analyses of first- and second-line randomized trials, is currently evaluating the amplification of *VEGFA* as a biomarker of long-term response to Ramucirumab-based treatment in patients with mGC to tailor treatment [13].

Consistent with this insight, our study aims to explore the role of VEGFA status as a predictive marker of the outcome of second-line therapy with Ramucirumab and PTX in patients with mGC. Towards this end, the *VEGFA* copy number, its expression in tumor tissues, and the density of tumor micro-vessels in mGC patient-derived samples were evaluated.

## 2. Materials and Methods

### 2.1. Patient Samples

In this retrospective study, collected samples were derived from 42 patients with advanced GC who were candidates for second-line treatment with Ramucirumab and PTX. The study was approved by the ethics committee (prot. n°139/c.e. 28-06-2017) and patients provided written informed consent for the collection of blood samples for biomarker analysis. Thirty-seven biopsy tissue specimens were provided by the Histopathology Unit of IRCCS “S. De Bellis” and five tissue samples were provided by the Surgery Division, University of Verona. The analyses were performed comparing two different groups of patients based on progression-free survival (PFS) defined as the time from random assignment in the study to disease progression or death. The first group of Rapid Progressor patients included those showing clinical or radiologic evidence of disease progression within the third month of therapy (PFS ≤ 3); the second group of Control Disease patients included those showing disease stability or response to treatment after the third month of therapy (PFS > 3). Patients in this group were followed up with three-monthly radiological controls until disease progression. All the results were correlated with the clinical data of each patient.

### 2.2. FISH Analysis

The commercially available ZytoLight SPEC *VEGFA*/CEN 6 Dual Color Probe (ZytoVision, Milan, Italy) was used to hybridize patient formalin-fixed paraffin-embedded (FFPE) sections mounted on microscope slides. Specifically, ZyGreen-labelled polynucleotides (excitation 503 nm/emission 528 nm), targeting sequences mapped in 6p21 (chr6: 43,633,271–43,985,142), including the *VEGFA* gene region, and ZyOrange-labelled polynucleotides (excitation 547 nm/emission 572 nm), targeting sequences mapped in 6p11.1-q11, specific to the alpha-satellite centromeric region D6ZZ on chromosome 6. FISH experiments were performed using the ZytoLight FISH-Tissue Implementation kit (ZytoVision, Milan, Italy) according to the manufacturer’s instructions. In particular, the sections were digested with Pepsin Solution for 8 min at 37 °C. For each case, the hybridization area, enriched in tumor cells, was identified by comparison with a slide with Hematoxylin and Eosin (H&E) staining [24].

Hybridized slides were observed using a Leica DM5500B fluorescence microscope (Leica Microsystems, Wetzlar, Germany), and the acquired images were analyzed with the CytoVision v.7.4 analysis software (Leica Microsystems, Wetzlar, Germany). For each case, at least 20 non-overlapping nuclei were analyzed. The *VEGFA* copy number was evaluated as reported in https://doi.org/10.3390/ph15060651 accessed on 1 June 2023 [25].

#### FISH Quantification

Two experienced pathologists independently analyzed a minimum of 20 tumor cell nuclei. FISH results were based on: (1) Chr6 copy number or (2) *VEGFA* gene/chr6 copy number ratio. Samples with a VEGFA/chr6 ratio of less than 2 were considered non-amplified; those with a ratio equal to or greater than 2 were considered amplified. This procedure followed 2018 ASCO/CAP guidelines for HER2 amplification in breast cancer [26]. Chr6 polysomy was defined as an average of Chr6 copy number.

### 2.3. IHC Analysis

FFPE tissue blocks were obtained from 42 patients subdivided into two groups: the Control Disease group (*n* = 28 patients) and the Rapid Progression group (*n* = 14 patients), including surgical specimens or biopsy samples.

Sections stained with hematoxylin and eosin (H&E) were reviewed by a pathologist to confirm the adequacy of the samples and to evaluate their morphologic and/or pathological characteristics. For IHC detection, 4 µm sections were cut and mounted on Apex Bond IHC slides (Leica Biosystems, Buffalo Grove, IL, USA). IHC staining procedures were performed on the BOND III automated immunostainer (Leica Biosystems, Buffalo Grove, IL, USA), from deparaffinization to counterstaining with hematoxylin using the Bond Polymer Refine Detection Kit (Leica Biosystems, Buffalo Grove, IL, USA). Tissue sections were incubated with two different antibodies: an anti-VEGFA primary antibody (MAb JH1121, Invitrogen, Waltham, MA, USA, 1:200 dilution) and an anti-CD34 primary antibody (Mab Qbend-10, Dako Agilent, Santa Clara, CA, USA, pre-diluted) for 30 min at room temperature. Antigen retrieval was performed by using BOND Epitope Retrieval Solution 2, a ready-to-use EDTA-based pH9 solution. Samples were recorded as negative when the number of stained cells was less than 5%.

#### 2.3.1. Scoring of VEGFA Staining

Scoring of 42 GC tissue sections was assessed by using a semi-quantitative pathology histoscore (H-score), defined as a method combining both percentages of positive-expression cells in the tissue slice and immunostaining intensities (hereinafter referred to as IHC-score). The IHC score was based on the membranous and cytoplasmic staining intensity level of VEGFA from 0 to 1+ (weak), 2+ (intermediate), or 3+ (strong). In brief, the H-score was calculated according to the formula: (0 × percentage of immunonegative cells) + (1 × percentage of weakly stained cells) + (2 × percentage of intermediately stained cells) + (3 × percentage of strongly stained cells). Thus, the H-score ranged from 0 (a tissue sample that is completely negative) to a maximum of 300 (a tissue sample in which all the cells show a 3+ staining), separating more distinctively the samples with predominantly high staining intensity from samples with a predominantly low staining intensity [27]. Samples with values ranging from 90 to 180 were considered to have high staining intensity. Those with values of 10 to 80 were considered to have moderate staining intensity and those with values below 10 were considered to have low staining intensity. All samples of this study were assessed by two pathologists working independently. In case of discrepancies in the assessments, the sections were discussed to reach a final agreement.

#### 2.3.2. Micro-Vessel Density (MVD) Quantification

To evaluate micro-vessel density (MVD), CD34 was used as a blood-vessel marker. MVD was assessed on the total cohort of patients *n* = 42. Samples with values greater than 100 were considered high density. Those with values between 50 and 100 were considered moderate density, and those with values below 50 were considered low density. Staining evaluation and vessel counting (number of vessels per field—0.56 mm^2^) were performed by two expert pathologists in a blinded manner [12].

### 2.4. Statistical Analysis

Patients’ data are reported as mean and standard deviation (M ± SD) for continuous variables and as frequency and percentages (%) for categorical variables. To test the association between the independent groups (Rapid Progression vs. Control Disease), the Fisher test was used for categorical variables, while the Wilcoxon Rank Mann-Whitney was used for continuous variables.

The Spearman rank correlation coefficient was used to test the strength and direction of the association existing between the two examined variables (i.e., between IHC VEGFA and CD34 in total, Control Disease, and Rapid Progression cohort and between IHC VEGFA and PFS, and IHC VEGFA and PFS).

The p-value is the result of statistical inference expressing the statistical probability of obtaining a certain result. To evaluate this value, the significance level was set to test the null hypothesis of no association; this two-tailed probability level was set at 0.05. The analyses were conducted with STATA software (StataCorp. 2023. Stata Statistical Software: Release 18, College Station, TX, USA) while RStudio (“Mountain Hydrangea” Release, Boston, MA, USA) was used for the plots.

## 3. Results

### 3.1. Gene Amplification of VEGFA in Tumor Tissue

*VEGFA* gene *status* was investigated on FFPE biopsy tissue slices using the VEGFA gene-specific probe (in green) and a chr6 centromeric probe (CEN6) as reference (in red). The analysis considered a total of 42 patients, 28 of whom had a PFS rate > 3 months in the Control Disease group and 14 with a PFS rate ≤ 3 months in the Rapid Progression group. The mean PFS of the entire population was 6.34 months. The PFS of the Control Disease group was 8.44 months, and the PFS of the Rapid Progression group was 2.28 months. No significant difference in age or gender was found between the two groups (Table 1).

Table 1 shows the overall percentage of patients with *VEGFA* copy number variation (amplification and polysomy) of Chr6 in each of the two groups analyzed. It was observed that *VEGFA* was amplified in 14.29% of all patients enrolled and, unexpectedly, in each cohort of the Control Disease group and Rapid Progression group. The percentage of polysomy was 23.81%, higher than amplification; although, the difference between the two groups was slight (21.43% Control Disease group vs. 28.57% Rapid Progression group). According to these results, copy number alteration of the *VEGFA* gene could not represent a predictive marker of response to therapy with Ramucirumab and PTX.

In the Control Disease group, 4 samples displayed *VEGFA* amplification, 1 was high grade with a *VEGFA/CEN6* ratio of 6.3 and a mean *VEGFA* copy number of 10 (Figure 1a), 2 had a ratio of 3.1 and 3, respectively, with a *VEGFA* mean copy number ≥6, and 1 showed a *VEGFA* mean copy number of 4 with a ratio of 2. Six patients were polysomic, (Figure 1b). The eighteen remaining patients showed a diploid state of VEGFA (Figure 1c).

Among the 14 samples from patients in the Rapid Progression group, 2 samples exhibited amplification: one with *VEGFA*/CEN6 ratio 2 and a *VEGFA* mean copy number of 4 and the other was the only sample of the whole casuistic with contextually high *VEGFA* mean copy number of about 10 and polysomy due to a Chr6 mean copy number of 4. (Figure 2a). Polysomy was present in 4 samples (Figure 2b) while the other 8 samples had diploid *VEGFA* status (Figure 2c).

### 3.2. Overexpression of VEGFA in Tumor Tissues

To investigate the expression of VEGFA in tumor tissues, IHC experiments were performed on tissue slices from the same block as those slices for H&E staining and FISH. Analyses were performed on the samples derived from the 28 patients in the Control Disease group and the 14 derived from the patients in the Rapid Progression group. Representative images are reported in Figure 1d–f and Figure 2d–f.

In the Control Disease group, all patients who presented *VEGFA* amplification had a high IHC VEGFA score between 90 and 180; patients with polysomy of Chr6 had a VEGFA expression score ranging between 40 and 160, and in one case, no overexpression was detected. In the 18 samples that did not have an altered *VEGFA* gene copy number, induction of the protein in tumor cells was of varying degrees: high overexpression in 6 cases (score 100–180), moderate overexpression in 6 (score 10–90), and absent in 5 patients.

In the Rapid Progression group, both patients with amplification did not show overexpression of the protein; similarly, among the 4 polysomy cases, one showed moderate overexpression with a score of 60, and 3 showed no induction of the protein; overexpression of the protein was found in the 3 diploid patients for the *VEGFA* gene, with a score ranging from 40 to 160.

Table 1 reports the overall results where the percentage of VEGFA overexpression is 70.83% in Control Disease patients, significantly higher (*p* = 0.02) than in Rapid Progression patients (28.57%). These differences were even more significant (*p* = 0.005) when considering the percentages of patients with high overexpression scores (90–180) where the percentages in the two groups were 58.33% and 7.14%, respectively.

Table 2 shows the comparative analysis between the two groups of patients in which the concurrence of combined variables in the analyzed samples was considered. Referring to the combination of gene copy number variation of *VEGFA* and its overexpression in the tumor, it was noteworthy that in the Control Disease group, the two variables are combined in 37.5% of cases compared with 7.14% observed in the Rapid Progression group. However, it should be emphasized that VEGFA overexpression is not always concomitant with the change in gene copy number and this was the case in both groups of patients.

### 3.3. Micro-Vessel Density in Tumor Tissues

To evaluate whether VEGFA amplification and/or overexpression increased vessel density, IHC analysis was assessed using CD34 staining as a marker of blood micro-vessels. The analysis was performed on the entire cohort of patients, and representative pictures for each patient group considered are shown in Figure 1g–i and Figure 2g–i. The samples analyzed were divided into three categories according to the density of the micro-vessels. Samples with values greater than 100 were considered high density, those with values between 50 and 100 were considered moderate density, and those with values below 50 were considered low density.

The category comprising a high density of micro-vessels was found in 25% of the Control Disease group and this percentage dropped to 14.25% in Rapid Progression patients. An even more significant difference between the two groups was shown in the intermediate category with CD34 values between 50 and 100 (moderate density), where the percentages of patients were 60.71% and 21.43%, respectively. The largest percentage of patients in the Rapid Progression group fell into the category with a low density of micro-vessels, between 1 and 50, where that percentage was 64.29% compared with 14.29% of patients in the Control Disease group (Table 1).

Micro-vessel density was highly variable, and there was no concomitance between *VEGFA* amplification/polysomy and high micro-vessel density. Instead, a concomitance between higher levels of protein expression and higher micro-vessel density could be reported as shown in Table 2.

### 3.4. Comparative Analysis between VEGFA and CD34 Expression

In the Control Disease group, the comparative analysis evidenced that protein overexpression (VEGFA score 40–180) was combined with high micro-vessel density (CD34 score > 50) in the high percentage of cases (66.67%), while only a small percentage of cases with high VEGFA scores were associated with a CD34 score below 50 (4.17%).

By contrast, in the Rapid Progression group, most of the cases showed a combination of no overexpression and low micro-vessel density (57.14%). The Spearman Rho correlation between these two parameters in the total cohort and in the two patients’ groups was calculated and reported in the graphs of Figure 3. The correlation indexes (ρ) indicated a moderate and significant correlation between the two parameters in each patient group.

### 3.5. VEGFA Expression and Micro-Vessel Density Correlate with Therapy Outcome

Finally, the correlation between the investigated parameters and the efficacy of Ramucirumab and PTX was evaluated. A significant and moderate correlation was found only for the parameters concerning the expression of VEGFA in tumor tissue (r = 0.50, *p* = 0.001) and micro-vessel density (r = 0.59, *p* = 0.0001), which was plotted in the graphs of Figure 4 reporting the respective correlation indices.

## 4. Discussion

Gastric cancers are widespread worldwide; nevertheless, there are few targeted therapies, and survival rates for these tumors are still poor [1,2,3]. Gaining an insight into the molecular features may facilitate a more rational classification of these tumors and provide a rationale for new therapies.

The monoclonal antibody Ramucirumab targeting the VEGFR2 receptor represents a milestone in the second-line treatment of mGC as monotherapy or in combination with PTX [4,5]. VEGF/VEGFR pathways are regulated by multiple ligands and receptors. This redundancy suggests that angiogenesis is a physiologically essential process that is likely to be conserved against any interfering effect of this pathway. Thus, this complex regulation by multiple factors may be implicated in the acquired resistance of tumors to angiogenic inhibitors [28]. Although about half of patients do not benefit from this anti-angiogenic drug, no predictive biomarkers have been identified to date.

VEGFA growth factor is a pivotal driver in tumor-mediated angiogenesis and is also important in sustaining tumor growth and metastasis formation [8]. Therefore, many studies have attributed negative prognostic significance to its induction in tumor tissues [29,30]; although, other reports have disputed this association [31]. Fuchs and colleagues hypothesized a positive predictive value of higher expression levels of VEGFR2 in tumor tissues of patients treated with Ramucirumab [30].

Several genomic characterization studies have pointed out that, although with a low incidence, amplification of the *VEGFA* gene locus is detectable in several cancer types, such as HCC, osteosarcoma, retinoblastoma, Merkel cell carcinoma, and carcinosarcoma and absent in healthy counterparts [12,32]. In addition, some types of cancer, such as breast or esophageal carcinoma, despite the absence of gene amplification were characterized by chr6 polysomy. However, the impact of polysomy on cancer and its role as a diagnostic marker is debatable [33,34].

In TCGA, *VEGFA* amplification was found in 7% of GC, almost exclusively in the chromosomal instable (CIN) subtype. Squamous cell carcinomas showed frequent genic amplification of *CCND1* and *SOX2* and/or *TP63*, while *ERBB2*, *VEGFA*, *GATA4*, and *GATA6* were more commonly amplified in adenocarcinomas. Esophageal adenocarcinomas strongly resemble the CIN variant of gastric adenocarcinoma, suggesting that these tumors could be considered a single disease entity with some molecular differences [35]. Therefore, it was hypothesized that *VEGFA* amplification in tumor cells could stimulate not only angiogenesis but also autocrine/paracrine stimulation of tumor growth, thus identifying a subgroup of patients with good response to anti-angiogenesis drugs. The prospective VERA study recruited patients with mGC undergoing second-line therapy with Ramucirumab alone or in combination with PTX with the aim to investigate the relationship between *VEGFA* amplification and clinical outcomes. Although the study met its primary endpoint, revealing 15% of tumors with *VEGFA* amplification in patients with higher PFS, validation analyses are needed to confirm *VEGFA* amplification as a biomarker of long-term response to Ramucirumab treatment in patients with mGC [13].

Therefore, further studies specifically addressing this issue are highly warranted. Furthermore, the need to correlate *VEGFA* gene status (e.g., amplification or polysomy) with VEGFA protein levels has emerged, given the broad understanding that gene amplification may not be the main mechanism behind protein overexpression. The extreme heterogeneity of tumors as well as transcriptional activation by other genes or post-transcriptional events may explain this discrepancy. Conversely, gene amplification does not automatically result in the overexpression of the corresponding protein; even in this case, the heterogeneity of the tumor plays a key role in determining the correspondence between the two events. In some extremely heterogeneous tumors, such as GC, studies in this regard are discordant and the only studies leading to guidelines concern HER2 [16,17]. The predictive role of other gene alterations, including *VEGFA*, remains an open question to date.

It has also been speculated that *VEGFA* amplification in some tumor types could be associated with increased micro-vessel density due to overexpression of VEGFA protein and that the density status of micro-vessels could modulate the response to anti-angiogenic therapy [12,18,36].

The present retrospective study extended a previous analysis focusing on the determination of serum levels of a panel of angiogenesis-related markers in a cohort of GC patients undergoing second-line therapy with Ramucirumab and PTX partly overlapping with that involved here. The results showed that pre-treatment levels of circulating factors, including members of the VEGF family (VEGFA, VEGFC, VEGFD, PLGF, VEGFR1, VEGFR2, and VEGFR3) were not a predictive marker of therapy outcome. Nevertheless, a greater decrease in circulating VEGFC and angiopoietin 2 levels correlated with an improved therapy outcome [37]. Based on these results, we extended the search for possible predictive markers by assessing the copy number status of the *VEGFA* gene as well as its expression in tumor tissue. Specifically, the study considered the status of the chromosome 6 locus in *VEGFA* gene maps and its expression levels in tumor tissues in addition to an analysis of micro-vessel density status.

The analysis enrolled 42 patients with a mean age of 65 years and mostly male (73.8%), receiving second-line therapy with Ramucirumab and PTX after fluoropyrimidine and platinum-based treatment failure. As in the previous study, the study population was divided into two groups based on clinical and radiologic responses to treatment after three months. The first group included 28 Control Disease patients (PFS > 3 months) and the second group included 14 Rapid Progressor patients (PFS ≤ 3 months) [37]. The mean PFS of the entire population was 6.34 months. The PFS of the Control Disease group was 8.44 months, and the PFS of the Rapid Progressor group was 2.28 months. No significant difference in age or gender was found between the two groups.

The status of the chromosome 6 locus in *VEGFA* maps was investigated by FISH, a method already validated for this type of analysis [12]. The results indicated a 14% rate of gene amplification and a higher rate (23.81%) of polysomy in all patients. Surprisingly, no significant differences in the frequency of both gene alterations were observed between the two groups of patients.

Furthermore, protein expression was scored on the same histological specimens. The expression score, considering both the percentage of positive cells and the expression intensity, was significantly higher (score 81.78) in patients with Control Disease than in Rapid Progressor patients (score 23.57). Moreover, a higher percentage of patients with longer PFS (58.33%) had a higher range of VEGFA expression (score 90–180). By contrast, only 7.14% of patients with lower PFS presented high expression scores of VEGFA, while the large majority (71.43%) exhibited no overexpression of the protein in the tumor. Thus, the key finding was that a significant concurrence of *VEGFA* amplification/polysomy and its overexpression in the tumor was observed in Control Disease patients, which was not found in the Rapid Progressor patients, where gene amplification/polysomy did not correspond to protein-increased expression in the tumor. According to these results, it can be assumed that VEGFA overexpression in tumor tissues could be considered an emerging biomarker to predict response to second-line treatment with Ramucirumab and PTX and to identify GC patients who will benefit from this therapy.

These findings become even more interesting if we consider the density of the micro-vessels detected by the expression score of its specific marker CD34. Most Control Disease patients (60.71%) exhibited CD34 values between 50 and 100 with a mean value of 103.96 (high density) in contrast to Rapid Progressors, where 64.29% of patients presented values between 1 and 50 with a mean value of 44.86 (low density). Furthermore, 66.67% of Control Disease patients have high VEGFA expression scores and a concurrent high density of micro-vessels. On the other hand, 57% of Rapid Progression patients have both normal VEGFA expression levels and low micro-vessel density.

Correlation analysis indicates a moderate and significant correlation between the two parameters concerning VEGFA expression and micro-vessel density in the entire cohort of patients and the Control Disease group. Finally, considering all patients, these two parameters were moderately and significantly associated with clinical outcomes expressed as PFS.

Overall, the results presented in this study suggest that the amplification/polysomy of the growth factor *VEGFA* is not a sufficient condition to determine the response to anti-angiogenic therapy with Ramucirumab, rather it is VEGFA protein overexpression in the tumor tissues associated with an induction of micro-vessels that leads to the therapeutic response in patients otherwise characterized by a poor prognosis.

It is widely accepted that overexpression of VEGFA is involved in the formation of vessels characterized by a completely impaired and not properly functional architecture. An impaired vasculature leads to hypoxic conditions resulting in the production of signal-activating molecules that create a microenvironment with an immunosuppressive phenotype devoid of effectors T cells. Thus, VEGF can be regarded not only as a major player in the angiogenic switch but also as a powerful immune modulator [11,38].

An altered status of VEGFA points to increased activation of downstream signal pathways involved in angiogenesis, tumor growth, and modulation of immune cells, resulting in greater responsiveness to the Ramucirumab. This drug is characterized by a higher affinity for the VEGFR2 receptor compared to VEGFA, thus explaining the greater efficacy of Ramucirumab-based therapy in patients showing concomitant amplification and overexpression of VEGFA. In patients with reduced VEGFA expression, other pathways relevant to regulating angiogenesis and tumor growth could be activated, explaining the resistance of these patients to Ramucirumab treatment. These alternative pathways could also be activated following Ramucirumab treatment, explaining the acquired resistance after the initial phase of a successful response to therapy. Therefore, a broader examination of angiogenic factors would be useful in studying predictive/prognostic markers of angiogenesis inhibitor therapies in mGC.

The relevance of this study was to identify VEGFA expression and/or micro-vessel density as possible predictive markers. Therefore, testing VEGFA and/or CD34 expression could be of high relevance in the clinical practice of mGC patients as this method is cost-effective and easy to perform on biopsy samples. Routine immunohistochemistry analyses for these two markers could allow for the selection of a cohort of patients eligible for anti-angiogenic therapy whose effects would be observed not only in achieving a window of “vessel normalization” but even in a shift of the immune phenotype in a permissive way [37,38,39]. These insights provide the rationale for either anticipating anti-angiogenesis drugs to frontline therapy or combining them with immunotherapy. In these frameworks, it is worth expanding this study to a greater number of patients to confirm the predictive value of VEGFA overexpression and/or micro-vessel density in GC patients receiving Ramucirumab-based therapy. A large cohort of patients would allow for the evaluation of the predictive value of VEGFA/micro-vessel density on a further stratification of the disease control group, restricting the analysis to long-term responders to Ramucirumab and PTX.

## 5. Conclusions

In the era of personalized oncology care, selecting patients for targeted therapies often requires the assessment of the status of a given biomarker, which could be assessed through both the analysis of the gene’s copy number, measured by FISH, and the evaluation of its protein overexpression, measured by IHC. The concordance between these two parameters is still hotly debated. In highly heterogeneous tumors, such as GC, studies addressing this issue are controversial, and *HER2* amplification is an important reference. The predictive role of other gene amplifications, including *VEGFA*, is still an open question. Data obtained in this study show that *VEGFA* amplification concomitantly with VEGFA overexpression and overexpression of VEGFA with micro-vessel density correlated with positive outcomes in patients treated with anti-angiogenic therapy plus PTX. Further research is warranted to further corroborate the use of VEGFA overexpression as a potential biomarker for the prediction of GC treatments. A large cohort of patients would allow for the evaluation of the predictive value of VEGFA/micro-vessel density on a further stratification of the Disease Control group, restricting the analysis to long-term responders to Ramucirumab and PTX.

## Figures and Tables

**Figure 1 biomedicines-11-02721-f001:**
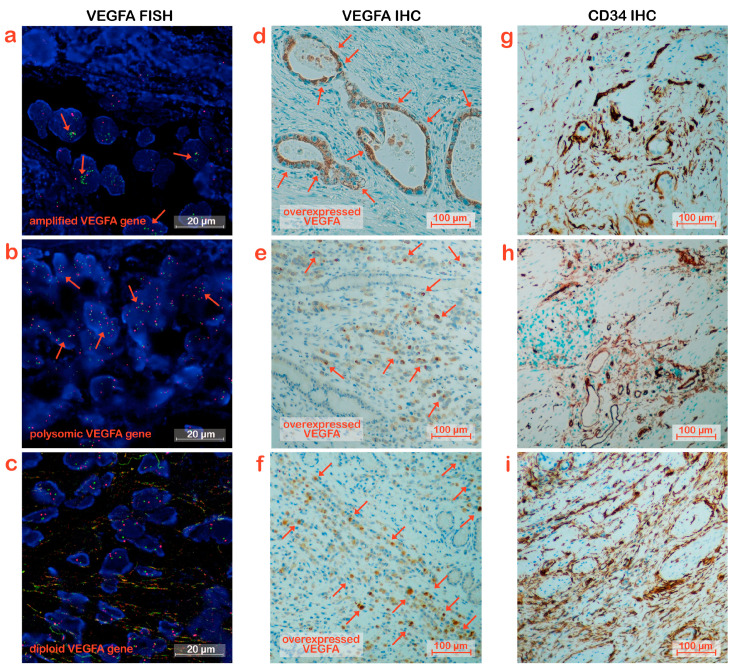
VEGFA expression and micro-vessel density in Control Disease GC patients. (**a**–**c**) Visualization of *VEGFA* gene locus status by FISH. (**a**) represents gene amplification, (**b**) polysomy and (**c**) diploid status. The green and red signals correspond to the *VEGFA* and reference centromere 6 gene region, respectively. (**d**–**f**) VEGFA immunostaining in tumor tissues. The three images all depicted VEGFA’s overexpression. Representative tumor cells overexpressing VEGFA were indicated by red arrows. (**g**–**i**) Micro-vessel density in tumor tissues evaluated by CD34 immunostaining. The three images all showed CD34′s overexpression.

**Figure 2 biomedicines-11-02721-f002:**
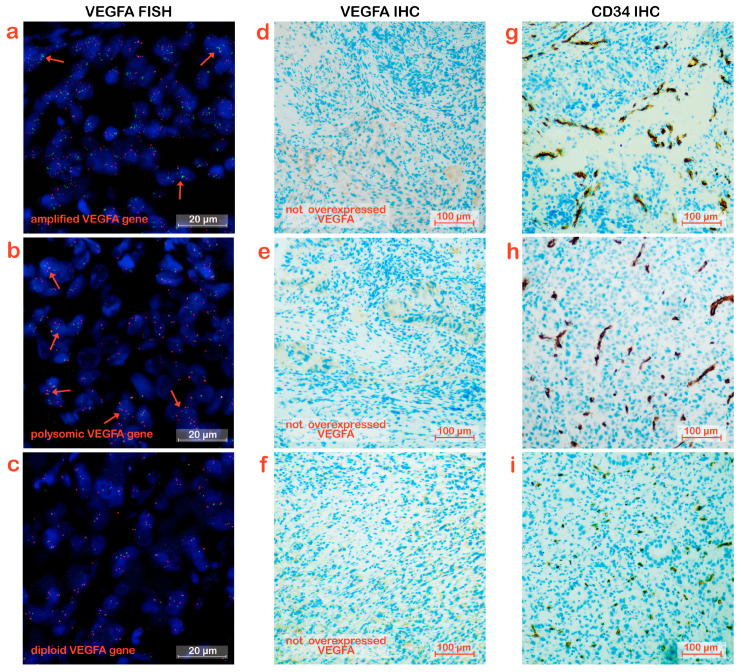
VEGFA expression and micro-vessel density in Rapid Progression GC patients. (**a**–**c**) Visualization of *VEGFA* locus status by FISH. (**a**) gene amplification, (**b**) polysomy and (**c**) the diploid status, were represented (Examples of nuclei with VEGFA copy numbers >2 are indicated by the red arrows). The green and red signals correspond to the *VEGFA* and reference centromere 6 gene region, respectively. (**d**–**f**) VEGFA immunostaining in tumor tissues. The three images depicted tumor cells not overexpressing VEGFA. (**g**–**i**) Micro-vessel density in tumor tissues evaluated by CD34 immunostaining. The three images showed low CD34 overexpression.

**Figure 3 biomedicines-11-02721-f003:**
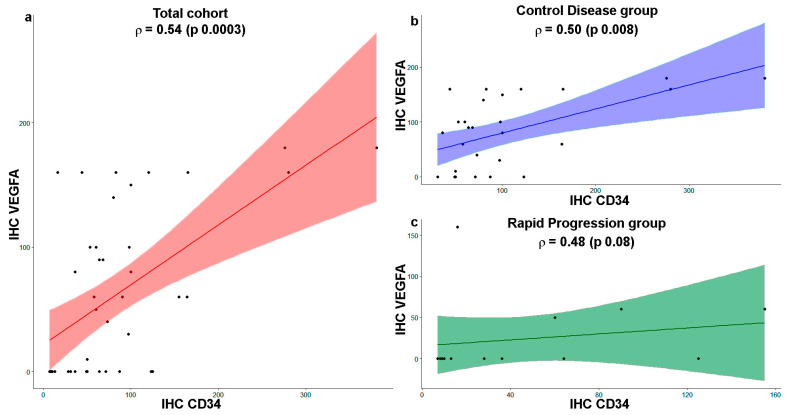
Correlation plot with single data (dots) between tissue VEGFA and CD34 expression. Spearman Rho correlation between IHC VEGFA and IHC CD34 scores in the total patient cohort in red (**a**) and in Control Disease in blue (**b**) and Rapid Progression in green (**c**) groups.

**Figure 4 biomedicines-11-02721-f004:**
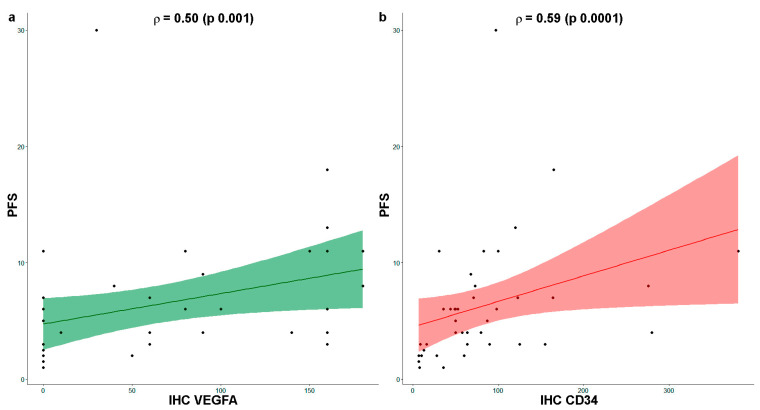
Correlation plot with single data (dots) between VEGFA or CD34 expression and progression-free survival (PFS). Spearman’s Rho correlation between PFS and IHC VEGFA score in green (**a**) and IHC CD34 score in red (**b**) in the total cohort of mGC patients.

**Table 1 biomedicines-11-02721-t001:** Comparison between Control Disease and Rapid Progression groups (*n* = 42).

Parameters *	Total Cohort (*n =* 42)	Group		*p* ^^^
		Control Disease (*n =* 28)	Rapid Progression (*n =* 14)	
Age (yrs)	64.93 ± 9.25	64.36 ± 9.84	66.07 ± 8.15	0.69
Gender (M) (%)	31 (73.81)	20 (71.43)	11 (78.57)	0.72 ^ψ^
FISH (%)				0.89 ^ψ^
Not Amplified	26 (61.90)	18 (64.29)	8 (57.14)	
Polysomic	10 (23.81)	6 (21.43)	4 (28.57)	
Amplified	6 (14.29)	4 (14.29)	2 (14.29)	
PFS (months)	6.34 ± 5.32	8.44 ± 5.46	2.28 ± 0.75	<0.0001
PFS (months) (%)				
≤3	14 (33.33)			
>3	28 (66.67)			
IHC VEGFA score	62.38 ± 65.44	81.78 ± 65.66	23.57 ± 46.01	0.003
IHC VEGFA score (%)				0.005 ^ψ^
0 (low)	17 (44.74)	7 (29.17)	10 (71.43)	
10–80 (moderate)	6 (15.79)	3 (12.50)	3 (21.43)	
90–180 (high)	15 (39.47)	14 (58.33)	1 (7.14)	
IHC VEGFA score (%)				0.02 ^ψ^
0	17 (44.74)	7 (29.17)	10 (71.43)	
10–180	21 (55.26)	17 (70.83)	4 (28.57)	
CD34	84.26 ± 77.51	103.96 ± 82.46	44.86 ± 48.14	0.003
CD34 (count/field) (%)				0.005 ^ψ^
1–50 (low density)	13 (30.95)	4 (14.29)	9 (64.29)	
50–100 (moderate density)	20 (47.62)	17 (60.71)	3 (21.43)	
>100 (high density)	9 (21.43)	7 (25.00)	2 (14.29)	

* As Mean and Standard Deviation (M ± SD) for continuous and percentage (%) for categorical variables. ^^^ Wilcoxon rank-sum test (Mann-Whitney), ^ψ^ Fisher’s test. Abbreviations: FISH, fluorescence in situ hybridization; PFS, progression-free survival; VEGFA, anti-human vascular endothelial growth factor.

**Table 2 biomedicines-11-02721-t002:** Comparison of combined variables between the Control Disease and Rapid Progression groups.

Parameters *	Total Cohort (*n* = 42)	Group		*p* ^^^
		Control (*n =* 28)	RP (*n =* 14)	
Combo				
VEGFA IHC and *VEGFA* FISH (%)				0.03
VEGFA IHC (0) and *VEGFA* Not Amplified	11 (28.95)	6 (25.00)	5 (35.71)	
VEGFA IHC (40–180) and *VEGFA* Not Amplified	11 (28.95)	8 (33.33)	3 (21.43)	
VEGFA IHC (0) and *VEGFA* Copy Number Variation	6 (15.79)	1 (4.17)	5 (35.71)	
VEGFA IHC (40–180) and *VEGFA* Copy Number Variation	10 (26.32)	9 (37.50)	1 (7.14)	
VEGFA IHC and CD34 IHC (%)				0.003
VEGFA IHC (0) and CD34 IHC (1–50)	10 (26.32)	2 (8.33)	8 (57.14)	
VEGFA IHC (40–180) and CD34 IHC (1–50)	2 (5.26)	1 (4.17)	1 (7.14)	
VEGFA IHC (0) and CD34 IHC (>50)	7 (18.42)	5 (20.83)	2 (14.29)	
VEGFA IHC (40–180) and CD34 IHC (>50)	19 (50.00)	16 (66.67)	3 (21.43)	

* As percentage (%) for categorical variables. ^^^ Fisher’s test. Abbreviations: FISH, fluorescence in situ hybridization; VEGFA, anti-human vascular endothelial growth factor; IHC, immunohistochemistry.

## Data Availability

Data are unavailable due to privacy or ethical restrictions.

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
