# Peer review of "VEGFA Status as a Predictive Marker of Therapy Outcome in Metastatic Gastric Cancer Patients Following Ramucirumab-Based Treatment"

_biomedicines, 2023, doi:10.3390/biomedicines11102721_

Round 1

Reviewer 1 Report

This study evaluated the role of VEGFA as a biomarker in the second line therapy with ramucirumab and paclitaxel in metastatic gastric cancer. This work is topical and I only have some minor comments:

1.       L47: I think this is the graphical abstract but there is no label and caption on it.

2.       Introduction: It is good to explain the rationale of VEGF further.

3.       L88: What is “[Error! Bookmark 88 not defined.]”? Similar issues in L111, L114, L200, L349, L380, L397, and L455.

4.       L136-146: Please provide a reference for the mentioned study.

5.       Please mention how to perform the error analyses (i.e. calculation of the p-values).

6.       Figure 3 and 4 are suggested to provide a color plot rather than black and white.

7.       Conclusion: Please mention what will be the future work in this study.

Reviewer 2 Report

This retrospective study assesses the potential role of VEGFA status as a predictor of response to therapy for metastatic gastric cancer patients with second-line treatment with Ramucirumab and PTX. The study is well-conducted, the methods are appropriately used, and the results sustain the conclusions.

Please provide how the follow-up of the patients was made. What was the median follow-up time for the entire and each group of patients? How was the PFS assessed?

A few paragraphs appear not to be adequately cited/ referenced. Please correct.

In Table 1, PFS is not measured in %. Please correct.  

The Figure from the beginning of the manuscript has no legend and is not cited in the text.

Please discuss how the present study's data would add value to clinical decision-making.

 Minor editing of English language required

Reviewer 3 Report

Topic of manuscript is suitable of for the biomedicines. VEGF play important part in the tumour development and result of clinical trials are interesting and reader and potentially useable in the therapy design. Nevertheless, their discussion is not sufficient. Obtained data (higher VEGF production and lesser microvessels density) could suggest, that role of hypoxia in the metastatic cancer.

Section about ELISA analysis of VEGF serum level (showed in the graphical abstract) should be included into method and result.

Some references (such as line 88) was not included correctly.

Round 2

Reviewer 3 Report

I have no serious guestions.

Some typos  (e.g. in the line 89) should be corected.
